# Interpreting Indirect Answers to Yes-No Questions in Multiple Languages

**Zijie Wang**[1][*]  **Md Mosharaf Hossain**[2][†]  **Shivam Mathur**[3][‡]  **Terry Cruz Melo**[1]
**Kadir Bulut Ozler**[1]  **Keun Hee Park**[4]  **Jacob Quintero**[1]  **MohammadHossein Rezaei**[1]
**Shreya Nupur Shakya**[1]  **Md Nayem Uddin**[4]  **Eduardo Blanco**[1][*]

[1]University of Arizona [2]Amazon [3]Walmart Global Tech [4]Arizona State University
`zijiewang@arizona.edu`

## Abstract

Yes-no questions expect a *yes* or *no* for an answer, but people often skip polar keywords. Instead, they answer with long explanations that must be interpreted. In this paper, we focus on this challenging problem and release new benchmarks in eight languages. We present a distant supervision approach to collect training data. We also demonstrate that direct answers (i.e., with polar keywords) are useful to train models to interpret indirect answers (i.e., without polar keywords). Experimental results demonstrate that monolingual fine-tuning is beneficial if training data can be obtained via distant supervision for the language of interest (5 languages). Additionally, we show that cross-lingual fine-tuning is always beneficial (8 languages).

## 1 Introduction

Multilingual Question-Answering has recently received substantial attention (Ruder and Sil, 2021; Shi et al., 2022). State-of-the-art models such as XLM-E (Chi et al., 2022), however, achieve only 68% to 76% F1-score on multilingual Question-Answering benchmarks such as MLQA (Lewis et al., 2020) and XQuAD (Artetxe et al., 2020). Large Language Models (LLMs) such as Instruct-GPT (Ouyang et al., 2022) obtain promising results with several English Question-Answering benchmarks (Rajpurkar et al., 2016; Choi et al., 2018; Lin et al., 2022). Closed-source, proprietary LLMs for which the training data is unknown raise issues regarding replicability and data leakage (Carlini et al., 2021). Open-source LLMs such as LLaMa are pretrained on Latin or Cyrillic scripts (Touvron et al., 2023) thus have limitations with other scripts.

| | |
|---|---|
| en | Q: [. . . ] Does it still feel like that?
A: I never really felt it was that way. [. . . ]
Interpretation: *No* |
| hi | Q: इस बार के बजट से क्या आप सहमत हैं?
(Are you satisfied with this budget?)
A: बेहतर बजट है अर्थव्यवस्था को मजबूती मिलेगी।
(This is a better budget, the economy will be strengthened.)
Interpretation: *Yes* |
| tr | Q: Teoman'ın konserine gidecek misin?
(Are you going to Teoman's concert?)
A: Patronum gelecek ayki planların ne olduğunu henüz söylemedi.
(My boss hasn't told me next month's plan.)
Interpretation: *Middle* |

Table 1: Yes-no questions and answers in English (en), Hindi (hi) and Turkish (tr). Answers do not include a *yes* or *no* keyword, but their interpretations are clear.

Yes-no questions are questions that expect a *yes* or *no* for an answer. Humans, however, often answer these kinds of questions without using a *yes* or *no* keyword. Rather, they provide indirect answers that must be interpreted to reveal the underlying meaning (see examples in Table 1). Indirect answers are used to ask follow-up questions or provide explanations for negative answers (Thompson, 1986), prevent wrong interpretations (Hirschberg, 1985), or show politeness (Brown and Levinson, 1978). This is true at least in English and the eight additional languages we work with. Note that question answering is usually defined as finding an answer to a question given a collection of documents. On the other hand, interpreting indirect answers to yes-no questions is defined as mapping a known answer to its correct interpretation.

Many NLP problems were initially investigated in English (Kann et al., 2019). Even though yes-no questions and indirect answers have been studied

---

[*]All authors except the first and last authors are listed in alphabetical order.

[†]The work does not relate to the position at Amazon.

[‡]Work done at Arizona State University.

for decades (Hockey et al., 1997; Green and Carberry, 1999), previous efforts to date have predominantly focused on English (Section 2). In this paper, we tackle this challenging problem in eight additional languages: Hindi (hi), Korean (ko), Chinese (zh), Bangla (bn), Turkish (tr), Spanish (es), Nepali (ne), and Persian (fa).

This paper focuses on multilingual interpretation of indirect answers to yes-no questions. Doing so opens the door to several applications. For example, dialogue systems could avoid inconsistencies and contradictions (Nie et al., 2021; Li et al., 2022). Consider the examples in Table 1. Follow-up turns such as *How long have you felt like that?*, *What else is required for your support?*, and *I can't wait to see you at the concert* (one per example) would be puzzling and probably frustrating to hear.

The main contributions are as follows:[1]

1. A distant supervision approach to collect yes-no questions and *direct* answers along with their interpretations. We use this approach to collect training data in five languages.

2. Evaluation benchmarks in eight languages in which no resources exist for interpreting *indirect* answers to yes-no questions.

3. Experimental results showing that training with the *direct* answers obtained via distant supervision is beneficial to interpret *indirect* answers in the same language.

4. Experimental results expanding on (3) and showing that multilingual training is beneficial, even when no additional training data for the language of interest is available.

## 2    Related Work

**Question Answering** Researchers have targeted, among others, factual questions (Morales et al., 2016), why questions (Lal et al., 2022), questions in context (Choi et al., 2018), questions over procedural texts (Tandon et al., 2019), and natural questions submitted to a search engine (Kwiatkowski et al., 2019). The problem is defined as finding answers to a given question (often within a set of documents). Unlike this line of work, interpreting indirect answers to yes-no questions is about determining the underlying meaning of answers—not finding them.

---

[1]Code, benchmarks, and multilingual training data obtained via distant supervision available at `https://github.com/wang-zijie/yn-question-multilingual`

**Yes-No Questions** have also been studied for decades (Hockey et al., 1997; Green and Carberry, 1999). Recent work includes large corpora such as BoolQ (Clark et al., 2019) (16,000 yes-no questions submitted to a search engine) and extensions including unanswerable questions (Sulem et al., 2022). Yes-no questions have also been studied within the dialogue domain (Choi et al., 2018; Reddy et al., 2019). These dialogues, however, are synthetic and constrained to a handful of topics. Circa (Louis et al., 2020), Friends-QIA (Damgaard et al., 2021), and SWDA-IA (Sanagavarapu et al., 2022), also explore yes-no questions in dialogues (crowdsourced, modified TV scripts, and phone conversations). We are inspired by their interpretations and use their corpora in our experiments (Section 5). All computational works on yes-no questions to date are in English. We are the first to target eight additional languages. Crucially, we do by exploring distant supervision and cross-lingual learning; our approach does not require tedious manual annotations.

**Multilingual Pretraining and Learning** Several efforts have investigated multilingual language model pretraining. Both mBERT (Devlin et al., 2019) and XLM-RoBERTa (Conneau et al., 2020) are masked language models pretrained on multilingual corpora. XLM-Align (Chi et al., 2021) improves cross-lingual domain adaptation using self-labeled word alignment.

Previous works have also focused on better fine-tuning approaches for cross-lingual transfer (Joty et al., 2017; Zheng et al., 2021). Others use machine translations to create synthetic multilingual training data or translate multilingual test benchmarks into English (Artetxe et al., 2023). In addition, several efforts have been made on multilingual Question Answering tasks. Wu et al. (2022) present a siamese semantic disentanglement model for multilingual machine reading comprehension. Riabi et al. (2021) propose a multilingual QA dataset leveraging question generation models to generate synthetic instances. Other work develops a Spanish SQuAD dataset based on a Translate-Align-Retrieve method (Carrino et al., 2020).

In this paper, we avoid translations. Instead, we use distant supervision. Our approach (a) is conceptually simple, (b) only requires unannotated corpora and rules for yes-no questions and direct answers in the languages of interest, and, importantly, (c) obtains statistically significantly better results than cross-lingual learning from English.

|  | Circa | SWDA-IA | Friends-QIA |
|---|---|---|---|
| # yes-no qs. | 34,268 | 2,544 | 5,930 |
| context? | no | yes | no |
| % Interpret. | | | |
| Yes | 56.0 | 61.9 | 48.8 |
| No | 37.5 | 23.2 | 24.6 |
| Middle | 2.8 | 14.9 | 26.6 |

Table 2: Statistics of existing English corpora with yes-no questions, indirect answers, and their interpretations.

## 3 Obtaining Training Data

For training purposes, we work with six languages. Specifically, we work with existing English corpora (Section 3.1) and corpora obtained in five additional languages via distant supervision (Section 3.2). As we shall see, multilingual transfer learning is successful even when no examples in the language of interest are available (Section 5.3).

### 3.1 Existing English Corpora

There are three English corpora that include yes-no questions, indirect answers, and their interpretations: Circa (Louis et al., 2020), SWDA-IA (Sanagavarapu et al., 2022) and Friends-QIA (Damgaard et al., 2021). Table 2 presents statistics. *Context* refers to the text around the question and indirect answer (i.e., dialogue turns before and after).

Circa was created by asking crowdworkers to write 34k yes-no questions that fit 9 scenarios (e.g., friends talking about food) and indirect answers. It does not include context. We note that the frequency of *Middle* interpretation is much lower than the other two interpretations. SWDA-IA includes 2.5k yes-no questions and indirect answers from SWDA (Stolcke et al., 2000), a telephone conversation dataset. It includes context (three turns before and after). Unlike Circa, questions and answers come from transcriptions of (almost) unconstrained conversations. Friends-QIA includes 5.9k yes-no questions and indirect answers derived from Friends, a TV show. Unlike Circa and SWDA-IA, questions and answers in Friends-QIA were manually modified to facilitate the task. For example, they remove some *yes* and *no* keywords in answers (e.g., *yes*, *yeah*, *yep*) but not others (e.g., *of course*, *absolutely*). Further, they add relevant information from the context to questions and delete interjections (e.g., *Hey!*) among others.

The three datasets do not consider the same interpretations (Circa: 8 options, SWDA-IA: 5 options, Friends-QIA: 6 options). We cluster them into *Yes*, *No* and *Middle* following our definitions (Section 4, Appendix A) for comparison purposes. Because Circa and SWDA-IA are the only corpora with "naturally occurring" questions and answers, we choose to not work with Friends-QIA.

### 3.2 Distant Supervision for New Languages

We follow a distant supervision approach to collect multilingual training data for interpreting indirect answers to yes-no questions. The only requirements are (a) (relatively) large unannotated corpora in the languages of interest and (b) rules to identify yes-no questions, direct answers, and their interpretations in each language. We found that native speakers can write robust rules after few iterations.
**Source Corpora** We made an effort to identify relevant corpora for the eight languages we work with but could only do so in five languages. The other three languages (Bangla, Nepali, and Persian) are spoken by millions, and we could certainly find digital texts in these languages. But (a) creating a large collection of dialogues and (b) splitting noisy transcripts into turns and sentences are outside the scope of this paper. Further, doing so would raise copyright and ethical considerations.

For Chinese, we select (a) NaturalConv (Wang et al., 2021), a synthetic dialogue dataset written by crowdworkers covering several topics, and (b) LCCC-base (Wang et al., 2020), posts and replies extracted from Weibo, a Chinese social media platform. For Spanish, we choose CallFriend (Canavan and Zipperlen, 1996), a corpus consisting of unscripted telephone conversations. For Hindi, we collect questions and answers from Twitter using their API. For Korean, we select a question-answering corpus from AI Hub,[2] which includes civil complaints and replies from public organizations. For Turkish, we identify FAQ (i.e., Frequently Asked Questions) and CQA (i.e., Community Question Answering) datasets from MFAQ (De Bruyn et al., 2021). All of these corpora are used in accordance with their licenses.

These corpora are diverse not only in terms of language. Indeed, they include different genres (written forum discussions, dialogue transcripts, question-answer pairs, etc.) and domains (social media, informal conversations, etc).
**Identifying Yes-No Questions** We define rules based on lexical matching to identify yes-no ques-

---

[2] www.aihub.or.kr

|  | Hindi | Korean | Chinese | Turkish | Spanish |
|---|---|---|---|---|---|
| # total turns | n/a | n/a | 7,220k | 4,264k | 64k |
| context? | no | no | yes | no | yes |
| # yes-no questions identified | 7,637 | 23,457 | 213,018 | 509,265 | 2,941 |
|    precision (random sample of 200) | 1.00 | 0.89 | 0.99 | 0.99 | 0.98 |
| # with indirect answers | 4,528 | 10,481 | 135,890 | 233,533 | 1,452 |
| # with direct answers | 3,109 | 12,976 | 77,128 | 275,732 | 1,489 |
|    precision (random sample of 200) | 0.65 | 0.96 | 0.93 | 0.97 | 0.93 |

Table 3: Statistics and evaluation of the rules used for distant supervision. The rules to identify yes-no questions are almost always correct. The ratio of indirect answers varies across languages, but it is always high (lowest: 44.7% in Korean; highest: 64.8% in Chinese). The rules to interpret direct answers are also very precise (0.93–0.97) except in Hindi (0.65), most likely because we work with Hindi tweets. We use the interpretations of *direct* answers obtained via distant supervision in these *five* languages to automatically interpret *indirect* answers in *eight* languages.

tions from the aforementioned corpora in each language. The Hindi rules are as follows. A tweet contains a yes-no question if it:

- contains a question mark and any of these bi-grams: क्या आप (do you), क्या हम (do we), क्या यह (will this), क्या कभी (does this, does this ever), यह हो सकता है (can this happen);
- does not contain these words: कहां (where), क्यो (why), कैसे (how), कौन (who), किसका (whose), कौनसा (which), या (or), कब (when);
- has between 3 and 100 tokens; and
- does not (a) contain links, @mentions, #hash-tags, or numbers or (b) come from unverified users, retweets, or replies to tweets.

**Identifying Direct Answers** The next set of rules identifies which yes-no questions are followed by a direct answer. We have rules that identify direct answers and their interpretations based on *yes* and *no* keywords. We complement these rules with rules to discard some answers that cannot be reliably interpreted regardless of keywords. The rules for Hindi are as follows. A reply tweet to a yes-no question is a direct answer if it:

- contains *yes* keywords: हां (Yes), हा (yes), हाँ (yes), जी (yes), जरूर (sure), सही (correct), निश्चित रूप (definitely), *yes, yeah, sure, of course, 100%*; or *no* keywords: नही (No), नहीं (no), मत (don't), न (not), *no, never, n't* (We include a few English keywords since people code-switch between these languages);
- does not contain links, #tags, or more than one @mention (i.e., only replies to the user who asked the yes-no question);
- does not contain question marks; and
- has between 6 to 30 tokens.

Appendix B details the rules for other languages.

**Analysis** The result of the rules for distant supervision consists of yes-no questions, direct answers, and (noisy) keyword-based interpretations. We prioritize precision over recall, as we need as little noise as possible for training purposes. We estimate quality with a sample of 200 instances per language (Table 3). The rules to identify yes-no questions are almost perfect across four out of five languages (Precision: 0.98–1.00). We identify thousands of yes-no questions in all the languages, and many of those (35.2%–65.3%) are followed by a direct answer that we can interpret with our rules. The ratio of *yes* and *no* interpretations varies across languages, and the precision of the rules to interpret direct answers is high (0.93–0.97) in all languages except Hindi (0.65). We believe that (a) the ratios of *yes* and *no* depend on the domain of the source corpora and (b) the low precision in Hindi is due to the fact that we work with Twitter as opposed to more formal texts. Note that all the question-answer pairs identified via distant supervision are interpreted with *yes* or *no*. Regardless of the quality of the data, whether it is useful in the training process is an empirical question (Section 5).

## 4 Benchmarks in New Languages

We are the first to work on interpreting answers to yes-no questions in languages other than English (Hindi, Korean, Chinese, Bangla, Turkish, Spanish, Nepali, and Persian). We set to work with questions and answers written in the languages of interest to avoid translationese (Koppel and Ordan, 2011), so we create new benchmarks with 300 question-answer pairs per dataset for each language (Chinese and Turkish: 600 samples; other languages: 300 samples). For the five languages

| | |
|---|---|
| [hi] Q: क्या हम सिर्फ़ धोखा खाने के लिए ही पैदा हुए हैं ? 
 (Are we born only to be cheated?) 
 A: दुर्भाग्यवश हम उस युग में पैदा हुए हैं। 
 (Unfortunately we are born in that era.) 
 Interpretation: *Yes* | [tr] Q: aşksız mutlu olabilir misiniz? 
 (Can you be happy without love?) 
 A: her şekilde mutlu olmasını bilirim. 
 (I know how to be happy anyway) 
 Interpretation: *Yes* |
| [ko] Q: 감기 기운이 있는 것 같은데 두통약 괜찮나요? 
 (I think I have a cold. Can I have a medicine for headache relief?) 
 A: 두통약보다는 그냥 감기약으로 드리겠습니다. 
 (I'll just give you cold medicine rather than a headache reliever.) 
 Interpretation: *No* | [es] Q: ¿Te ha seguido molestando él? 
 (Has he continued to bother you?) 
 A: A cada rato. 
 (All the time.) 
 Interpretation: *Yes* |
| [ne] Q: धर्म निरपेक्षता मान्नुहुन्छ ? 
 (Do you believe in secularism?) 
 A: संविधानको व्यवस्था हो। त्यसैले मान्नुपर्छ। 
 (It is a provision of the constitution. So it should be accepted.) 
 Interpretation: *Middle* | [zh] Q: 车里有矿泉水瓶吗? 
 (Any water bottle in the car?) 
 A: 还是用罐头瓶吧。 
 (Let's just use a can.) 
 Interpretation: *No* |
| [fa] Q: تو از بالا میبینی هوا تمیز شده؟ 
 (You are looking from a high elevation, is the weather clean?) 
 A: الان دارم دماوندمو میبینم قشنگ 
 (I am now clearly looking at the Damavand (Mountain).) 
 Interpretation: *Yes* | [bn] Q: আমার মাথায় একটু হাত দিয়ে দেখতো জ্বর আছে কিনা ? 
 (Can you please check if I have a fever?) 
 A: এখন কম। 
 (It's less now.) 
 Interpretation: *Yes* |

Table 4: Examples from the benchmarks we create in eight languages. Answers whose interpretations lean towards *yes* or *no* are annotated as such; *middle* is used for 50/50 splits and unresponsive answers.

for which there are large unannotated corpora and we have built rules for (Section 3), we select yes-no question-answer pairs *without* a direct answer (i.e., those identified by our rules as yes-no questions but not followed by a direct answer). We collect 300 question-answer pairs for the other languages from Bangla2B+ (Bhattacharjee et al., 2022, Bangla), Kantipur Daily (Nepali),[3] and LDC2019T11 (Mohammadi, 2019, Persian).

**Annotation Guidelines** We manually annotate the interpretations of indirect answers to yes-no questions in the eight languages using three labels:

- *Yes*: the answer leans towards yes, including *probably yes*, *yes under some conditions*, and strong affirmative answers (e.g., *Absolutely!*).
- *No*: the answer leans towards no.
- *Middle*: *Yes* or *No* interpretations do not apply.

Our interpretations are a coarser version than those used in previous work. Appendix A presents a mapping, and Appendix C details the guidelines.

Table 4 presents examples. In the Hindi (hi) example, the *yes* interpretation relies on the negative sentiment of the answer. In the Turkish (tr) example, the *yes* interpretation requires commonsense knowledge about fasting, which includes not drinking water. Similarly, the Persian (fa) example requires commonsense (seeing a mountain implies clear weather). In the Korean (ko) example, the answer provides an alternative, thereby rejecting the

---

[3] https://ekantipur.com/

request. In the Spanish (es) example, the answer affirms the inquiry without using *yes* or similar keywords. In the Nepali (ne) example, the answer is interpreted as *middle* since it mentions laws rather than discussing an opinion. In the Chinese (zh) example, the answer suggests using a can, implying that they do not have any water bottles in the car. In the Bangla (bn) example, the answer confirms that the questioner has a fever (despite it is lower).

**Inter-Annotator Agreements** Native speakers in each language performed the annotations. We were able to recruit at least two annotators for five languages (zh, hi, es, tr, and bn) and one annotator for the rest (ne, ko, fa). Inter-annotator agreements (linearly weighted Cohen's $\kappa$) are as follows: Turkish: 0.87, Hindi: 0.82, Spanish: 0.76, Bangla: 0.73, and Chinese: 0.67. These coefficients are considered substantial (0.6–0.8) or (nearly) perfect (>0.8) (Artstein and Poesio, 2008). We refer the reader to Appendix D for a detailed Data Statement.

**Dataset Statistics and Label Frequency** Table 5 presents the statistics of our training datasets and benchmarks in the new languages. Since we obtain the training datasets via distant supervision (Table 3), they only include yes-no questions with direct answers and thus their interpretations are limited to be *yes* or *no*. For the benchmarks, we select instances from each language randomly. This allows us to work with the *real* distribution of interpretations instead of artificially making it uniform.

|  | Hindi | Korean | Chinese | Bangla | Turkish | Spanish | Nepali | Persian |
|---|---|---|---|---|---|---|---|---|
| **Training instances** | | | | | | | | |
| # instances | 3,109 | 12,976 | 77,128 | n/a | 275,732 | 1,489 | n/a | n/a |
| % yes | 27.0 | 92.0 | 57.7 | n/a | 94.4 | 60.1 | n/a | n/a |
| % no | 73.0 | 8.0 | 42.3 | n/a | 5.6 | 39.1 | n/a | n/a |
| **Benchmarks (Test instances)** | | | | | | | | |
| # instances | 300 | 300 | 600 | 300 | 600 | 300 | 300 | 300 |
| % yes | 44.4 | 52.7 | 38.5 | 34.3 | 43.7 | 54.2 | 46.7 | 46.7 |
| % no | 43.3 | 27.3 | 30.5 | 32.0 | 32.7 | 12.0 | 36.3 | 32.0 |
| % middle | 12.7 | 20.0 | 31.0 | 33.7 | 23.6 | 32.8 | 17.0 | 21.3 |

Table 5: Number of instances and label frequency in the datasets obtained via distant supervision (top, five languages), and benchmarks for new languages (bottom, eight languages). *Yes* interpretations are almost always the most frequent in both training datasets and benchmarks. *Middle* is the least frequent in five languages in benchmarks.

|  | Hindi | Korean | Chinese | Bangla | Turkish | Spanish | Nepali | Persian |
|---|---|---|---|---|---|---|---|---|
| **Baselines** | | | | | | | | |
| Majority | 0.27 | 0.37 | 0.21 | 0.18 | 0.27 | 0.34 | 0.30 | 0.29 |
| Random | 0.35 | 0.34 | 0.34 | 0.31 | 0.33 | 0.36 | 0.37 | 0.34 |
| **XLM-RoBERTa trained with** | | | | | | | | |
| Circa | 0.42 | 0.58 | 0.34 | 0.42 | 0.54 | 0.45 | 0.51 | 0.55 |
| SWDA-IA | 0.29 | 0.56 | 0.34 | 0.23 | 0.47 | 0.36 | 0.51 | 0.47 |
| Circa + SWDA-IA | 0.33 | 0.60 | 0.36 | 0.45 | 0.55 | 0.43 | 0.53 | 0.55 |

Table 6: Results (F1) obtained with XLM-RoBERTa, a multilingual transformer, trained with English corpora (i.e., cross-lingual in the new language). Appendix E presents additional results (P and R). Training with Circa and SWDA-IA is more beneficial in most cases. Training with SWDA-IA alone, however, shows limited improvements.

As shown in Table 5 (bottom), *yes* is always the most frequent label (34.3%–54.2%), but the label distributions vary across languages.

## 5 Experiments

We develop multilingual models to reveal the underlying interpretations of indirect answers to yes-no questions. We follow three settings: (a) cross-lingual learning to the new language (after training with English, Section 5.1); (b) monolingual fine-tuning via distant supervision with the language of interest (Section 5.2); and (c) multilingual fine-tuning via distant supervision with *many* languages (Section 5.3). Following previous work (Section 2), we use the question and answer as inputs.

We conduct our experiments with two multilingual transformers: XLM-RoBERTa (Conneau et al., 2020) and XLM-Align (Chi et al., 2021). Both are obtained from HuggingFace (Wolf et al., 2020) and were pretrained on hundreds of languages. We split our benchmarks into validation (20%) and test (80%), and report results with the test split. Cross-lingual learning was conducted with a model

trained and validated exclusively in English (with Circa and SWDA-IA). Monolingual and multilingual fine-tuning use the validation split in each language for hyperparameter tuning, The test set is always the same. We will discuss results with XLM-RoBERTa, as it outperforms XLM-Align by a small margin. Appendix F and G detail the results with XLM-Align and the hyperparameters.

### 5.1 Cross-Lingual Learning from English

We start the experiments with the simplest setting: training with existing English corpora (Circa and SWDA-IA) and evaluating with our benchmarks. We consider this as a cross-lingual learning baseline since the model was neither fine-tuned nor validated with the new languages.

Table 6 presents the results. Training with Circa is always better than (or equal to) SWDA-IA, but combining both yields the best results with most languages. The only exception is Hindi, where training with Circa obtains better results. This is intuitive considering that Circa is a much larger dataset than SWDA-IA (Table 2). The lower re-

|  | Yes | | | No | | | Middle | | | All | | | |
|---|---|---|---|---|---|---|---|---|---|---|---|---|---|
|  | P | R | F | P | R | F | P | R | F | P | R | F | $\%\Delta\mathrm{F}^{en}$ |
| Hindi | 0.47 | 0.79 | 0.59 | 0.49 | 0.22 | 0.31 | 0.25 | 0.13 | 0.17 | 0.45 | 0.46 | 0.43 | 2 |
| Korean | 0.70 | 0.84 | 0.76 | 0.61 | 0.76 | 0.68 | 0.75 | 0.07 | 0.12 | 0.69 | 0.67 | 0.62 | 3 |
| Chinese | 0.47 | 0.81 | 0.59 | 0.53 | 0.56 | 0.54 | 0.92 | 0.08 | 0.14 | 0.63 | 0.50 | 0.43 | 19 |
| Turkish | 0.63 | 0.87 | 0.73 | 0.72 | 0.57 | 0.63 | 0.46 | 0.26 | 0.33 | 0.62 | 0.63 | 0.60 | 9 |
| Spanish | 0.56 | 0.66 | 0.60 | 0.21 | 0.48 | 0.29 | 0.71 | 0.20 | 0.31 | 0.57 | 0.47 | 0.46 | 2 |

Table 7: Results obtained in multiple languages with XLM-RoBERTa, a multilingual transformer, trained blending with English corpora (best combination from Table 6) and the instances obtained with distant supervision in the language we evaluate with. $\%\Delta\mathrm{F}^{en}$ indicates the improvement compared to training with English only (Table 6). Training with the new language is always beneficial and it only requires defining a handful of rules.

sults with SWDA-IA for Hindi are likely due to the mismatch of domains (SWDA-IA: phone conversation transcripts, Hindi: Twitter).

## 5.2 Fine-Tuning with the New Language

We continue the experiments by training models with English corpora (gold annotations, Circa and SWDA-IA) and the data obtained via distant supervision for the language of interest (noisy data, Section 3.2 and Table 5). We adopt the fine-tuning methodology by Shnarch et al. (2018) to blend training data from English corpora and the additional instances obtained via distant supervision. Briefly, we start the training process (first epoch) with the concatenation of the English data and data for the new language. Then, we reduce the data for the new language by a ratio $\alpha$ after each epoch. We choose the English corpora based on the best combination from Table 6 for each language. We found that reducing the data for the new language (as opposed to the English data) yields better results. We believe this is because the former: (a) is noisier (distant supervision vs. human annotations) and (b) does not include any *middle* interpretations.

Table 7 presents the results. Recall that additional data is only available in five languages. The results show that blended training with English and the new language is always beneficial. The improvements ($\%\Delta\mathrm{F}^{en}$) range from 2% to 19%. They are the largest with the languages for which we have the most additional data (Table 3): Turkish (9%) and Chinese (19%). Note that despite the additional Hindi data is noisy (0.67 Precision, Table 3), we observe improvements (2%).

## 5.3 Fine-Tuning with Several Languages

We close the experiments exploring whether it is beneficial to train with languages other than En-

glish and the language of interest. In other words, we answer the following questions: Does multilingual fine-tuning yield better results?

We use the same blending strategy as in monolingual fine-tuning (Section 5.2) but adopt a greedy approach to find the best combination of additional languages to train with. We start with the model trained with English and the data for the language of interest if available (Table 7). Otherwise, we start with the model trained with English (Table 6). Then, we add data from an additional language (one at a time) and select the best based on the results with the validation split. We continue this process until including additional languages does not yield higher results with the validation split.

Table 8 presents the results. The technique is successful. Compared to monolingual fine-tuning (Table 7), multilingual fine-tuning is always beneficial ($\%\Delta F = 2$–28). More importantly, the improvements with respect to cross-lingual learning (Table 6) are even higher across all languages ($\%\Delta F^{en} = 2$–53). For five out of eight languages, the improvements are at least 11% and statistically significant (McNemar's test, $p < 0.05$). Note that multilingual fine-tuning obtains significantly better results with two of the languages for which we could not find source corpora to use distant supervision: Nepali (17%) and Bangla (22%). We hypothesize that the low gains with Persian might be due to the fact that it is the only one using a right-to-left script.

**Which languages are worth training with?** Some of the language combinations that are worth blending with are surprising (Table 8). For example, Chinese (zh) is beneficial for Hindi (hi), Turkish (tr) and Spanish (es). This may be due to the fact that Chinese is one of the languages for which we collect the most data via distant supervision. Similarly, Turkish (tr) is useful for Korean (ko).

| Blending w/ | Yes | | | No | | | Middle | | | All | | | | |
|---|---|---|---|---|---|---|---|---|---|---|---|---|---|---|
| | P | R | F | P | R | F | P | R | F | P | R | F | $\%\Delta F^{en}$ | $\%\Delta F$ |
| Hindi  hi, zh, es | 0.53 | 0.66 | 0.59 | 0.51 | 0.50 | 0.51 | 0.25 | 0.03 | 0.06 | 0.49 | 0.52 | 0.49 | 16* | 14 |
| Korean  ko, tr | 0.68 | 0.88 | 0.77 | 0.65 | 0.67 | 0.66 | 0.83 | 0.11 | 0.20 | 0.70 | 0.68 | 0.63 | 5 | 2 |
| Chinese  zh, es, ko | 0.56 | 0.69 | 0.63 | 0.53 | 0.68 | 0.59 | 0.69 | 0.31 | 0.43 | 0.59 | 0.56 | 0.55 | 53* | 28* |
| Bangla  hi | 0.53 | 0.64 | 0.58 | 0.47 | 0.60 | 0.52 | 0.76 | 0.42 | 0.54 | 0.59 | 0.55 | 0.55 | 22* | n/a |
| Turkish  tr, zh | 0.65 | 0.83 | 0.73 | 0.69 | 0.69 | 0.69 | 0.41 | 0.18 | 0.25 | 0.61 | 0.64 | 0.61 | 11* | 2 |
| Spanish  es, zh, hi, ko | 0.55 | 0.71 | 0.62 | 0.26 | 0.32 | 0.29 | 0.58 | 0.29 | 0.38 | 0.52 | 0.51 | 0.49 | 9 | 6 |
| Nepali  tr, hi | 0.69 | 0.77 | 0.73 | 0.62 | 0.69 | 0.65 | 0.41 | 0.18 | 0.25 | 0.62 | 0.64 | 0.62 | 17* | n/a |
| Persian  hi | 0.60 | 0.80 | 0.69 | 0.53 | 0.58 | 0.55 | 0.73 | 0.19 | 0.31 | 0.61 | 0.59 | 0.56 | 2 | n/a |

Table 8: Results obtained with XLM-RoBERTa, a multilingual transformer, trained blending with English corpora (best combination from Table 5.1) and the instances obtained with distant supervision in other languages (Column 2). We show only the best combination of languages. $\%\Delta F^{en}$ and $\%\Delta F$ indicate the improvements compared to training with English only (best in Table 6) and blending English and the language we evaluate with (Table 7). An asterisk indicates that the improvements are statistically significant (McNemar's test (McNemar, 1947), $p < 0.05$).

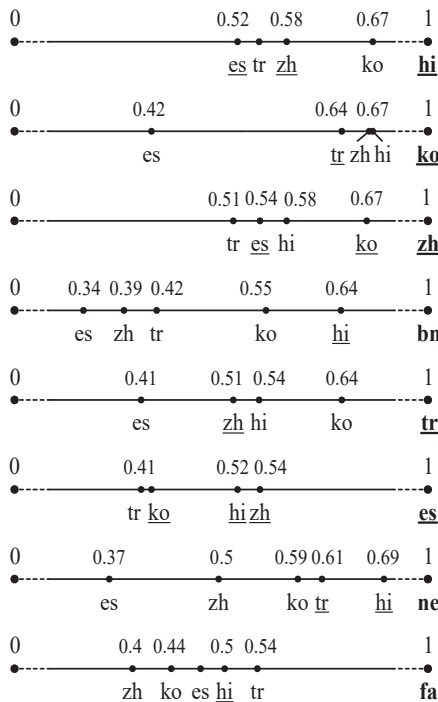

Figure 1: Language similarities between the eight languages we evaluate with (boldfaced) and each of the languages we use distant supervision with (hi, ko, zh, tr, and es). Underlining indicates that a language is worth blending with (Table 8). The more similar the languages, the more likely to be useful, although there are exceptions (e.g., Korean and either Turkish or Hindi).

In order to analyze these surprising results, we sort all language pairs by their similarity (Figure 1). We define language similarity as the cosine similarity between their lang2vec vectors (Littell et al., 2017), more specifically, we use syntactic and language family features from the URIEL typological database. Generally speaking, the more similar a language the more likely it is to be useful for multilingual fine-tuning. There are only a few exceptions (e.g., Korean with Hindi and Turkish).

## 5.4 Examples of Errors

Table 9 presents questions and answers for which our best model fails to identify the correct interpretation. In the Chinese (zh) example, the *no* interpretation could be obtained by contrasting *urban area* and *airport*. However, the model lacks such commonsense knowledge. Similarly, in the Bangla (bn) example, a wife being afraid of dogs most likely indicates that there are no dogs in the household. In the Korean (ko) example, the answer provides both *yes* and *no* for different days of the weekend, but the model appears to consider only Sunday. In the Spanish (es) example, the question has a negation thus the answer ought to be interpreted as *yes*. Answers in both the Turkish (tr) and Hindi (hi) examples include a polar distractor that confuses the model (tr: *no longer* yet the author has a favorite game; hi: *stop* yet the answer ought to be interpreted as *nobody can stop paying taxes*. The answer in Nepali (ne) indicates it could (or could not) happen while the model only takes it as affirmation. The answer in Persian (fa) is a rhetorical question with negation, thus the correct interpretation is *yes*.

## 5.5 A Note on Large language Models

Large Language Models (LLMs) obtain impressive results in many tasks (Mishra et al., 2022). Researchers have also shown that LLMs can solve problems even with malicious prompts (Webson and Pavlick, 2022), casting a shadow on what they

| Instance | Prediction | Gold |
|---|---|---|
| [zh] Q: 你住机场附近吗？ (Do you live near the airport?)
A: 我住市区西园饭店 (I live in the urban area, Xiyuan Hotel.) | *yes* | *no* |
| [ko] Q: 주말도 영업하시나요? (Do you open on weekends too?)
A: 주말은 토요일은 여섯시까지 영업하고, 일요일은 휴무입니다 (On weekends, we are open until 6pm on Saturday and closed on Sunday.) | *no* | *middle* |
| [es] Q: y no se lo ha operado todavía? (And you haven't had surgery yet?)
A: bueno cuando yo he estado Esta segunda vez (Well I've been here for the 2nd time.) | *no* | *yes* |
| [tr] Q: en sevdiğiniz bilgisayar veya mobil oyun var mı? (Do you have a favorite computer or mobile game?)
A: pc oyunlarına bağımlılığım kalmadı, lakin 3 senedir arada bir oynadığım dead by daylight var. (I am no longer addicted to PC games, but there is Dead by Daylight, which I have been playing occasionally for 3 years.) | *no* | *yes* |
| [ne] Q: सरकारमा गएपछि त्यो मुद्दा तह लगाउन सकिन्छ कि भन्ने चर्चा छ नि ? (Is there a discussion that after going to the government, the case can be dropped?)
A: त्यो हुन सक्छ ।(That could be.) | *yes* | *middle* |
| [fa] Q: کسی این کارو میکنه ؟ (Does anyone do this?)
A: نمیکنن ؟ (Don't they?) | *no* | *yes* |
| [hi] Q: क्या हम टैक्स देना बंद कर दें? (Should we stop paying taxes?)
A: सांस लेना भले बंद कर दो, पर टैक्स तो देना ही पड़ेगा ।(Even if you stop breathing, you will still have to pay taxes.) | *yes* | *no* |
| [bn] Q: আপানার বাড়িতে কুকুর আছে, স্যার? (Sir, do you have a dog in your house?)
A: আমার গিন্নি কুকুর ভয় পায়। (My wife is afraid of dogs.) | *yes* | *no* |

Table 9: Examples of error interpretations to instances from our benchmarks, made by our best model (Table 8).

may understand. LLMs do not outperform our best model (Table 8). First, we note that open-source LLMs are "pretrained only on Latin or Cyrillic scripts" (Touvron et al., 2023), thus they cannot process most of the languages we work with. Second, closed-source LLMs such as ChatGPT are likely to have been pretrained with the data in our benchmarks, so data leakage is an issue. We randomly selected 30 instances for each language from our benchmarks and fed them to ChatGPT via the online interface using the prompt in Appendix H. The best model for each language (Table 8) outperforms ChatGPT by 16.4% on average with these instances (F1: 54.25 vs. 63.13). Regardless of possible data leakage, we acknowledge that ChatGPT obtains impressive results in a zero-shot setting.

## 6   Conclusions

We have tackled for the first time the problem of interpreting indirect answers to yes-no questions in languages other than English. These kinds of answers do not include *yes* or *no* keywords. Their interpretations, however, are often either *Yes* or *No*. Indeed, *Middle* accounts for between 12.7% and

33.7% depending on the language (Table 3).

We have created new evaluation benchmarks in eight languages: Hindi, Korean, Chinese, Bangla, Turkish, Spanish, Nepali, and Persian. Additionally, we present a distant supervision approach to identify yes-no questions and direct answers. Training with this new, noisy, language-specific data is always beneficial—and significantly better in five languages. Cross-lingual learning from many languages to a new language is the best strategy. This is true even for the three languages for which distant supervision was not explored because of difficulties finding large unannotated corpora (Bangla, Nepali, and Persian). Our approach successfully learns to interpret *indirect* answers from *direct* answers obtained via distant supervision.

Our future work includes exploring more robust multilingual fine-tuning (Zheng et al., 2021). We also plan to explore applications of the fundamental research presented here. In particular, we are interested in improving dialogue consistency and avoiding inconsistencies by ensuring that generated turns are compatible with the correct interpretation of indirect answers to yes-no questions.

## Limitations

We double-annotate the benchmarks in five out of the eight languages. We are not able to do so in the other three languages (Bangla, Nepali, and Persian) since we are unable to recruit a second native speaker. While great care was taken, including single annotators double and triple checking their work, we acknowledge that not reporting inter-annotator agreements in these three languages is a weakness.

We adopt three labels (*Yes*, *No* and *Middle*) to represent the interpretations of answers to yes-no questions. Some previous works use finer-grained label sets as we discussed in Section 3.1. Considering that (a) there is no universal agreement about the possible ways to interpret answers to yes-no questions and (b) interpretations of answers other than the three we adopted are rare (less than 10% in Circa and Friends-QIA, and less than 20% in SWDA-IA), we argue that three labels are sound.

We run the experiments with two models: XLM-RoBERTa and XLM-Align. We acknowledge that there are other transformers such as InfoXLM and XLM-E. However, they either obtain similar (or worse) results on existing benchmarks, or are not open-source at the time of writing. Regarding open-source Large Language Models (LLMs) such as LLaMa and Alpaca, we do not run experiments on them since (a) fine-tuning with even the smallest version (parameters: 7B) requires significant computing resources; (b) they are pretrained only on Latin or Cyrillic scripts thus may have limitations targeting other languages.

Our in-context learning experiments are conducted with ChatGPT based on GPT-3.5 (*text-davinci-002*), which was the state-of-the-art model at the time we conducted the experiments. GPT-4 is already out, and better language models will continue to come out. We acknowledge that testing with GPT-4 may obtain better results, but we argue doing so is not necessary: (a) querying on GPT-4 based ChatGPT interface is limited to a few samples per hour, and (b) OpenAI has never announced the list of supported languages so we would be shooting in the dark. A more serious issue is the fact that GPT-X may have been trained with the data we work with, so any results are to be taken with a grain of salt.

We tune several hyperparameters (including the blending factor $\alpha$, listed in Table 12 and Table 13) with the train and development splits, and report results with the test set. The results are taken from the output of one run. We acknowledge that the average of multiple runs (e.g., 10) would be more reliable, but they also require much more computational resources (literally, 10 more times).

## Ethical Considerations

**Data sources and collection.** Our benchmarks use source texts in accordance with their licenses. We collect corpora in Spanish and Persian from Linguistic Data Consortium[4] under a non-commercial license. Samples in Turkish from MFAQ are used under the Apache-2.0 license. Samples in Chinese from NaturalConv and LCCC are used under a non-commercial license and the MIT license respectively. Samples in Hindi from Twitter are used under the Twitter developer policy. [5] Samples in Korean from AI Hub are used under a non-commercial license. Samples in Bangla from Bangla2B+ are used under the CC BY-NC-SA 4.0 license. Samples in Nepali from Kantipur Daily are used under a non-commercial license.

## Acknowledgments

We would like to thank Siyu Liu and Xiao Liu for their help in annotating the Chinese benchmark, and the Chameleon platform (Keahey et al., 2020) for providing computational resources. We also thank the reviewers for their insightful comments.

This material is based upon work supported by the National Science Foundation under Grant No. 1845757. Any opinions, findings, and conclusions or recommendations expressed in this material are those of the authors and do not necessarily reflect the views of the NSF.

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

## A Mapping Heterogeneous Interpretations to *Yes*, *No*, and *Middle*

The three English corpora (Circa, SWDA-IA, Friends-QIA) use compatible but different interpretations for yes-no questions (Section 3.1). Here we list the mapping to our three labels (*Yes*, *No*, and *Middle*). This mapping is straightforward given their definitions.

Circa (relaxed labels):
- *Yes → Yes*
- *No → No*
- *Yes, subject to some conditions → Yes*
- *In the middle, neither yes nor no → Middle*
- *Other*: discard
- *N/A*: discard

SWDA-IA:
- *Yes → Yes*
- *Probably Yes → Yes*
- *Middle → Middle*
- *Probably No → No*
- *No → No*

Friends-QIA:
- *Yes → Yes*
- *No → No*
- *Yes, subject to some conditions → Yes*
- *Neither yes nor no → Middle*
- *Other*: discard
- *N/A*: discard

## B Rules to Identify Yes-No Question and Direct Answers in Multiple languages

**Hindi**   In addition to the rules listed in Section 3.2, we remind the reader that Hindi data was collected before Twitter changed its policies on verified accounts and API usage.

**Turkish**   For the Turkish corpus (MRQA), we define the following rules to identify yes-no questions:

- The conversation turn contains any of the following keywords: *mıyım, miyim, muyum, müyüm, mısın, misin, musun, müsün, mı, mi, mu, mü, mıyız, miyiz, muyuz, müyüz, mısınız, misiniz, musunuz, müsünüz*. All of these words come from the same root (*mi, mı, mu, mü*), which is used to make a sentence a yes-no question. There is no direct translation into English.
- The turn does not contain wh-questions keywords: *ne (what), nerede (where), ne zaman (when), nasıl (how), nasil (informal how), neden (why), kim (who), hangi (which), and kimin (whose).*
- The turn is less than 50 tokens.

The rules to identify direct answers in Turkish rely on *yes* (*evet* (yes), *evt* (yes, informal), *eet* (yes, informal), *tabii* (of course), *tabi* (of course, informal), *tabiiki* (of course, informal), *tabiki* (of course, informal), *aynen* (absolutely), and *hıhı* (yes, informal)) and *no* (*hayır* (no), *hayir* (informal no), *hyr* (informal no), and *yoo* (informal no)) keywords.

Regardless of keywords, we discard question-answer pairs if the *is_accepted* field is set to *false*, which indicates the answer was not selected.

**Spanish**   For the Spanish corpus (CallFriend), we define the following rules to identify yes-no questions. A conversation turn includes a yes-no question if:

- it contains a verb;
- ends with a question mark ('?'); and
- does not contain the following words or phrases: *por que* (why), *cuando* (when), *donde* (where), *como* (how), *cuanto* (how much/many), *quien* (who), *cual* (which, singular), or *cuales* (which, plural).

The rules to identify direct answers in Spanish are defined as follows: The answer turn (i.e., the turn after the question turn) contains *yes* keywords (*si* (yes), *claro* (sure), *correct* (correcto), *vale* (ok), *por supuesto* (sure), *quizas* (maybe), *de acuerdo* (understood), *asi es* (that's right)) or *no* keywords (*no* (no), *nah*, *nope*, *no se* (I don't know), *no lo se* (I don't know), *no estoy seguro* (I am not sure), *ni idea* (no idea)).

In Spanish, we consider both the accented (and proper) spelling and the unaccented ones.

**Chinese**   For the Chinese corpora (NaturalConv and LCCC-base), we define the following rule to identify yes-no questions: A conversation turn ends with a modal particle 吗 or 嘛, followed by a question mark ('？').[6]

We also define rules to identify direct answers in the turn following yes-no questions. Our rules are defined as follows. The answer turn (i.e., the turn after the question turn) starts with:

- *yes* keywords ( 对 (right), 好 (okay), 嗯 (uh-huh), 恩 (uh-huh), 当然 (of course), 必须 (must)) or *no* keywords: (不 (no), and 没 (absence)) ; or
- the first verb in the question, either in the affirmative or negative form. For example, in the yes-no question 我可以坐这里吗？ (Can I sit here?), the first verb is 可以 (can). This rule matches turns that start with the verb 可以 (can) or its negated form 不可以 (cannot).

**Korean**   For the Korean corpus, we define the following rules to identify yes-no questions. Recall that the corpus contains questions and answers, so the rules are designed to differentiate between yes-no questions and other questions rather than identifying yes-no questions in any corpus. A turn contains a yes-no question if it does not:

- contain the following keywords indicating wh-questions: 어떻게 (how), 뭐가 (what), 뭐 가 (what is), 어떤 (which is), 무슨 (what), 뭐에 요 (what), 뭐예요 (what), 얼만 (how much), 들어가요 (how much available), 들어가나요 (how much do you need), 몇 번 (how many), 몇 평 (how big), 몇평 (how big), 몇 번 (how many times), 언제 (when), 어디 (where), 어 딧 (where to), 무엇을 (which one), 어떤 거 (which one), 걸려요 (how much time does it take), 몇 분 (how many minutes), 몇분 (how many minutes), 어떻게 (how), 몇 시 (what time), 차량번호확인 (what VIN number), 비 용은요 (how much cost), 어느 (where), 얼마 (how much), 어떤 부분 (which part), 왜 안 (why not), 어느 쪽 (which direction), 어떤 물품 (which item), 시까지 (until what time), 몇시까지 (by what time), 몇 일 (what date), 몇시 (what time), 몇일 (what date), 머 알아 야 (what should I know), 뭐 필요 (what do I need), 몇 대 (how many cars), 몇 대 (how many cars), 앞자리는요 (what are the digits); and

- end in the following modal particle words or phrases that indicate statements instead of questions, as in 버스 그 난폭운전에 대해 가지고 좀 불편신고를 좀 할려구요. (I was calling if I can file a civic complaint about reckless driving). The full list of particles is as follows: 할려구요, 하려구요, 아니면, 했는데요, 같, 은데요, 거든요, 렸는데요, 같아서요, 싫어서요, 은요, 가지구요, 같애 서, 하는데요, 카는데요, 좀 할께요, 가가 지고, 놔두고 내렸는데.

The rules to identify direct answers in Korean rely on keywords:

- *yes* keywords: 네 (yes), 예 (yeah), 그렇 (right), and 맞아 (correct) ;
- *no* keywords: 아 없 (ah no), 예 없 (yeah, no), 안타깝 (unfortunately), 아니(no), 아뇨 (ney), and 아닙 (no) .

## C   Annotation Guidelines for the Benchmarks

We conduct manual annotations to obtain ground truth interpretations (i.e., gold labels) for indirect answers. We work with three labels: *Yes*, *No*, and *Middle (Unknown)*, In order to minimize inconsistencies, we define labels as follows:

- *Yes*: The answer leans towards (or implies) yes or yes under certain conditions or constraints. The latter could be interpreted as *probably yes* (e.g., Q: Ever done this before? A: Once.).
- *No*: The answer leans towards (or implies) no, no under certain conditions or constraints (probably no), or provides arguments for no. The last two could be interpreted as *probably no* (e.g., Q: Can I at least have a drink? A: It's ten thirty in the morning.)
- *Middle (Unknown)*: The answer is unresponsive (e.g., changes the topic) or uninformative (e.g., "I don't know"). It should imply or lean towards neither *yes* nor *no*. (e.g., Q: Can you connect me to someone else? A: Well what's the situation?)

## D   Data Statement

As recommended by Bender and Friedman (2018), we provide a data statement to better understand the new data presented in this paper.

**Curation Rationale**

We develop new datasets to help interpret indirect answers to yes-no questions in eight languages.

---

[6] This is Chinese, not the English question mark ('?')

First, we adopt a rule-based approach to collect yes-no questions, *direct* answers, and interpretations of the answers in five languages. The interpretations were obtained by automatically mapping the positive keyword to *Yes*, and the negative keyword to *No*. This data is noisy (see quality estimation in Table 3; Precision = 0.93–0.97 except in Hindi (0.67)) and intended to be used for distant supervision.

Second, we collect yes-no questions with *indirect* answers and manually annotate their interpretations in the eight languages we work with. For the five languages we used distant supervision with, we collect questions followed by answers not identified by our rules (i.e., without polar keywords). For the other three languages, we collect questions and indirect answers from scratch.

Spanish and Chinese corpora come with context (i.e., the turn before the question and after the answer) since they were obtained from conversations. The corpora include only the questions and answers in the other languages.

We list all the original data sources for our data in Section 3.

**Language Variety**

We work with eight languages to develop our datasets. The language list is as follows: Bangla as spoken in Bangladesh (bn-BD), Chinese as spoken in Mainland China and written with simplified characters (zh-CN), Korean as spoken in South Korea (ko-KR), Turkish as spoken in Turkey (tr-TR), Spanish as spoken in United States, Canada, Puerto Rico, or The Dominican Republic (es-US, es-CA, es-PR, es-DO), Persian as spoken in Iran (fa-IR), and Nepali as spoken in Nepal (ne-NP). We are not able to provide language variety information for Hindi since the data comes from Twitter.

**Speaker Demographic**

Our corpora are collected from various sources. We are not able to provide their speaker demographic since such information is absent in the original sources. However, all the corpora are spoken (or written) by native speakers.

**Annotator Demographic**

We recruit 12 annotators consisting of two women and ten men. Their ages range from 18 to 30 years old. Among them, three individuals are native speakers of Chinese, two of Spanish, two of Bangla, one of Persian (and proficient in Turkish), one of Nepali (and proficient in Hindi), one of Hindi, one

of Turkish, and one of Korean. All of them are highly proficient in English. Ethnic backgrounds are as follows: four individuals are from East Asia, four are from South Asia, one is from the Middle East, one is from Europe, one is from North America, and one is from South America. Socioeconomic backgrounds are as follows: all annotators reported that they are middle class.

Educational backgrounds are as follows. We have two undergraduate, nine graduate students, and one with a doctoral degree. Nine of them work in NLP-related research areas, two work in other research areas within Computer Science, and one has Mathematics background.

**Speech Situation**

Chinese corpora are from two sources: (a) written by crowdsourcing workers who are given a specific topic, and (b) written by users on social media platforms. The Spanish corpus consists of transcripts of telephone conversations between humans. The Nepali corpus is written by journalists from a daily news website. The Korean corpus is written by civilians and covers complaints and replies from public organizations. The Hindi corpus is written by Twitter users. The Turkish corpus is written by humans and consists of frequently asked questions and community questions. The Bangla corpus comes from a crawled dataset from online sources. The Persian corpus is transcriptions of spoken language by humans having informal conversations, including telephone calls and face-to-face interactions.

# E   Detailed Results for Cross-Lingual Transferring to a New Language

We provide supplemental results in Table 10 that further include Precision (P), Recall (R) and F1-score (F) for our cross-lingual learning experiments. This table complements Table 6.

# F   Experimental Results with XLM-Align

We present the experimental results obtained with XLM-Align in Table 11. The results are listed in three blocks and each block is comparable to Table 6, Table 7 and Table 8 respectively. We observe a similar trend in the results with both models, while XLM-RoBERTa outperforms XLM-Align in most cases.

| | Hindi | | | Korean | | | Chinese | | | Bangla | | |
|---|---|---|---|---|---|---|---|---|---|---|---|---|
| | P | R | F | P | R | F | P | R | F | P | R | F |
| **Baselines** | | | | | | | | | | | | |
| Majority | 0.20 | 0.45 | 0.27 | 0.28 | 0.53 | 0.37 | 0.15 | 0.38 | 0.21 | 0.13 | 0.35 | 0.18 |
| Random | 0.38 | 0.33 | 0.35 | 0.39 | 0.33 | 0.34 | 0.34 | 0.34 | 0.34 | 0.31 | 0.31 | 0.31 |
| **XLM-RoBERTa trained with** | | | | | | | | | | | | |
| Circa | 0.51 | 0.50 | 0.42 | 0.61 | 0.64 | 0.58 | 0.35 | 0.43 | 0.34 | 0.51 | 0.47 | 0.42 |
| SWDA-IA | 0.49 | 0.45 | 0.29 | 0.54 | 0.64 | 0.56 | 0.30 | 0.44 | 0.34 | 0.22 | 0.34 | 0.23 |
| Circa + SWDA-IA | 0.40 | 0.44 | 0.33 | 0.67 | 0.65 | 0.60 | 0.45 | 0.44 | 0.36 | 0.55 | 0.47 | 0.45 |

| | Turkish | | | Spanish | | | Nepali | | | Persian | | |
|---|---|---|---|---|---|---|---|---|---|---|---|---|
| | P | R | F | P | R | F | P | R | F | P | R | F |
| **Baselines** | | | | | | | | | | | | |
| Majority | 0.19 | 0.44 | 0.27 | 0.25 | 0.51 | 0.34 | 0.21 | 0.47 | 0.30 | 0.21 | 0.46 | 0.29 |
| Random | 0.19 | 0.44 | 0.27 | 0.40 | 0.34 | 0.36 | 0.39 | 0.36 | 0.37 | 0.38 | 0.33 | 0.34 |
| **XLM-RoBERTa trained with** | | | | | | | | | | | | |
| Circa | 0.63 | 0.61 | 0.54 | 0.59 | 0.51 | 0.45 | 0.51 | 0.53 | 0.51 | 0.59 | 0.57 | 0.55 |
| SWDA-IA | 0.46 | 0.56 | 0.47 | 0.30 | 0.48 | 0.36 | 0.52 | 0.59 | 0.51 | 0.57 | 0.53 | 0.47 |
| Circa + SWDA-IA | 0.59 | 0.61 | 0.55 | 0.51 | 0.49 | 0.43 | 0.56 | 0.57 | 0.53 | 0.58 | 0.57 | 0.55 |

Table 10: Detailed Results obtained in multiple languages with XLM-RoBERTa, a multilingual transformer, trained with English corpora. This table complements Table 6 by including precision and recall values.

| | Hindi | Korean | Chinese | Bangla | Turkish | Spanish | Nepali | Persian |
|---|---|---|---|---|---|---|---|---|
| **XLM-Align trained with** | | | | | | | | |
| Circa | 0.40 | 0.57 | 0.43 | 0.48 | 0.54 | 0.43 | 0.56 | 0.50 |
| SWDA-IA | 0.27 | 0.37 | 0.21 | 0.17 | 0.27 | 0.34 | 0.42 | 0.42 |
| Circa + SWDA-IA | 0.37 | 0.54 | 0.42 | 0.49 | 0.51 | 0.46 | 0.56 | 0.51 |
| **XLM-Align trained with** | | | | | | | | |
| **English corpora + one new language** | | | | | | | | |
| (Table 7) | 0.37 | 0.57 | 0.52 | n/a | 0.61 | 0.43 | n/a | n/a |
| **XLM-Align trained with** | | | | | | | | |
| **English corpora + language combination** | | | | | | | | |
| (Table 8) | 0.40 | 0.58 | 0.54 | 0.51 | 0.56 | 0.53 | 0.60 | 0.50 |

Table 11: Results (F) obtained with XLM-Align, trained with three strategies (Section 5). For the third strategy (the third block), we adopt the same language(s) as we train with XLM-RoBERTa to blend with. Comparing to XLM-RoBERTa (Table 6, Table 7 and Table 8), most results are worse. However, both models obtain a similar trend.

# G Training Details

Referring to Section 5, we conduct our experiments on an off-the-shelf XLM-RoBERTa-base (parameters: 279M) and XLM-Align-base model (parameters: 279M) from HuggingFace (Wolf et al., 2020). Both models are multilingual transformers with attention mechanism which has been utilized in various domains (Liu et al., 2023; Mnih et al., 2014). We run the experiments on a single NVIDIA Tesla V100 (32GB) GPU. Depending on the size of training datasets, the training time may vary, but approx-

imately it takes 10 minutes to train 1 epoch.

We tune with several hyperparameters to obtain the experimental results with XLM-RoBERTa-base. We list them in Table 12. We also tune the blending factor $\alpha$ (i.e., the ratio of data from distant supervision added to each training epoch). We list the $\alpha$ for each benchmark in Table 13. For experiments with XLM-Align-base, we follow the same hyperparameters and blending factors.

|  | Table 6 | Table 7 | Table 8 |
|---|---|---|---|
| Maximum epochs | 10 | 30 | 30 |
| Warmup steps | 0.01 * training steps | 0.01 * training steps | 0.01 * training steps |
| Batch size | 32 | 16 | 16 |
| Optimizer | AdamW | AdamW | AdamW |
| Learning rate | 2e-5 | 2e-5 | 2e-5 |
| Weight decay | 1e-2 | 1e-2 | 1e-2 |
| Gradient clipping | 1.0 | 1.0 | 1.0 |

Table 12: Tuned hyperparameters for our experiments with XLM-RoBERTa-base model.

|  | Hindi | Korean | Chinese | Bangla | Turkish | Spanish | Nepali | Persian |
|---|---|---|---|---|---|---|---|---|
| Blending factor $\alpha$ | 0.2 | 0.5 | 0.2 | 0.2 | 0.2 | 0.2 | 0.2 | 0.2 |

Table 13: Tuned blending factor $\alpha$ for our experiments with XLM-RoBERTa-base model.

```
I need you to help me understand
the underlying meanings of
indirect answers to yes-no
questions.  You can choose
from Yes, No, or Middle.  You
should reply to me with Yes,
No or Middle based on your
interpretation of the answer.

Question:  "<Question from
benchmarks>"
Answer:  "<Answer from
benchmarks>"

Does the answer mean Yes, No or
Middle?
```

Figure 2: Prompt used with ChatGPT

# H    Details of In-context Learning with ChatGPT

We test 30 random samples for each language with ChatGPT (*text-davinci-002*) and our best model, as explained in Section 5.5. Figure 2 presents the prompt. In order to give ChatGPT the most credit possible, we manually map the generated answers from ChatGPT into *Yes*, *No*, and *Middle*.