# OpenReview forum: "Interpreting Indirect Answers to Yes-No Questions in Multiple Languages"
_EMNLP/2023/Conference — EMNLP 2023 Findings_

### Official Review · Reviewer_KQFS · 2023-07-25

**Soundness:** 3

**Excitement:**

2: Mediocre: This paper makes marginal contributions (vs non-contemporaneous work), so I would rather not see it in the conference.

**Paper Topic And Main Contributions:**

The paper introduces a new multilingual benchmark consisting of eight languages for interpreting indirect answers to yes/no questions. Among them, five languages have both training and evaluation data, while the remaining three languages are provided with evaluation data only. The data collection was achieved using distant supervision, primarily relying on lexical matching as the initial set of rules. The authors conducted experiments with various settings, including training with only English data or training with multiple languages while evaluating on unseen language. The findings indicate that combined training with multiple languages yields better results compared to using a single language for training. Additionally, training with direct answers improves the model's ability to interpret indirect answers.

**Reasons To Accept:**

* The benchmark covers multiple low-resource languages, broadening the evaluation scope to include diverse linguistic contexts.

* Native speakers were involved in writing the rules.

* Some of the shown examples require a form of knowledge or commonsense reasoning for a correct interpretation, making it more challenging.

**Reasons To Reject:**

* The experimental results are somewhat expected, as the addition of more training data from other languages typically leads to improved results.

* In general, the task is relatively simple.

* The settings where the models were trained on (Circa + SWDA-IA) could be better described as a cross-lingual setting as well instead of  "zero-shot" since they were already exposed to the task of interpreting answers to yes/no questions.

* Table 3 appears a bit confusing; it might be more beneficial to have separate tables for the final statistics of the benchmark and the evaluation of the rules to enhance clarity and readability.

**Reproducibility:**

4: Could mostly reproduce the results, but there may be some variation because of sample variance or minor variations in their interpretation of the protocol or method.

**Reviewer Confidence:**

4: Quite sure. I tried to check the important points carefully. It's unlikely, though conceivable, that I missed something that should affect my ratings.

---

> ### Author Rebuttal · Authors · 2023-08-28
>
> Thank you for the thoughtful review. We truly appreciate your comments.
>
> **Reasons To Reject**
> - **RE: The experimental results are somewhat expected, as the addition of more training data from other languages typically leads to improved results.**
>   - The statement is inaccurate according to our results (and thus perhaps the results are not somewhat expected). We observe that adding additional languages (i.e., more data) does not necessarily improve the results (Section 5.3, Table 8). We stop the greedy strategy when adding a new language is no longer beneficial. Our hypothesis is that such improvements are more likely correlated to language similarity (see l. 469 - 486).
>
> - **RE: In general, the task is relatively simple.**
>   - The task is simple to formulate, but it is not simple to solve. Supervised models obtained modest results (Tables 5-8, most F1-scores are below 0.60), and ChatGPT does worse (Section 5.4).
>
>  - **RE: The settings where the models were trained on (Circa + SWDA-IA) could be better described as a cross-lingual setting as well instead of "zero-shot" since they were already exposed to the task of interpreting answers to yes/no questions.**
>    - Thank you for the rewording suggestion. We will change the word accordingly in the next version of the paper. “crosslingual” is indeed the correct term.
>
> - **RE: Table 3 appears a bit confusing; it might be more beneficial to have separate tables for the final statistics of the benchmark and the evaluation of the rules to enhance clarity and readability.**
>    - Thank you for the suggestion. We will have a new block to differentiate the statistics and evaluation of the rules, or have two tables if there is enough space once we can use an extra page.

---

### Official Review · Reviewer_jtqq · 2023-08-04

**Soundness:** 3

**Excitement:**

4: Strong: This paper deepens the understanding of some phenomenon or lowers the barriers to an existing research direction.

**Paper Topic And Main Contributions:**

This paper presents a yes-no questions resource with their interpretation (yes, no, middle), over 8 languages: Hindi, Korean, Chinese, Bangla, Turkish, Spanish, Nepali, and Persian. The majority of the samples in the dataset were collected based on applying rules on corpora in that language, but there were also annotation efforts on 300 samples per 8 languages.

XLM-RoBERTa is trained on multiple different settings (English 0-shot, train on one language, train on a combination of multiple languages) and results are reported and compared. The authors also show which languages are most helpful if used during training for another language.

**Questions For The Authors:**

See above.

**Reasons To Accept:**

- Great yes-no Q resource covering multiple languages.
- Most of the data was automatically collected, but there were annotation efforts on 300 samples per 8 languages.
- Many experimental results are reported comparing the 0-shot setting with finetuning on one language and finetuning on multiple languages.
- Interesting analyses on including which languages in training data can help get better results on another language.

**Reasons To Reject:**

- I'm a bit confused about the experiments and final results. There are two sets of data, one which is extracted based on some rules (is noisy and questions have direct answers), and one set is annotated. I'm not sure how this data is combined in experiments. Is it combined across all splits, or the test set only includes annotated data? If distant supervision data is included in test set, then the answers are direct and easier to classify. IMO, the results should have been broken down into annotated samples, and distant supervision data. The main goal is interpreting "indirect Answers", and I can't tell how successful the approach and different training setups are based on current reported results. If the test set only included annotated data, please make that more clear in the writeup.
- Multiple experiments were done, but there is a lack of analysis and explanation of the results. Scores are not very high which is ok, but a subsection explaining what were some of the weaknesses/strengths of the model, with some examples, would have made the paper stronger IMO.

**Reproducibility:**

4: Could mostly reproduce the results, but there may be some variation because of sample variance or minor variations in their interpretation of the protocol or method.

**Reviewer Confidence:**

2: Willing to defend my evaluation, but it is fairly likely that I missed some details, didn't understand some central points, or can't be sure about the novelty of the work.

---

> ### Author Rebuttal · Authors · 2023-08-28
>
> Thank you for the thoughtful review. We truly appreciate your comments.
>
> **Reasons to Reject**
> - **RE: I'm a bit confused about the experiments and final results. [...] I'm not sure how this data is combined in experiments. Is it combined across all splits, or the test set only includes annotated data? If distant supervision data is included in test set, then the answers are direct and easier to classify.**
>     - As explained in l. 194, the instances obtained with distant supervision are only used for training. The test set only includes the instances from benchmarks that were annotated manually (l. 377). There is no overlap between the instances in the training set (i.e., obtained with distant supervision) and the test set; the selection criteria (l. 308) guarantees so. We will add further explanations to make this more clear.
> - **RE: Multiple experiments were done, but there is a lack of analysis and explanation of the results. Scores are not very high which is ok, but a subsection explaining what were some of the weaknesses/strengths of the model, with some examples, would have made the paper stronger IMO.**
>    - Thank you for the suggestion. We will include the predictions of our models for each language in the public repository and include examples of errors for each language in the appendices.

---

### Official Review · Reviewer_DJgs · 2023-08-11

**Soundness:** 3

**Excitement:**

4: Strong: This paper deepens the understanding of some phenomenon or lowers the barriers to an existing research direction.

**Paper Topic And Main Contributions:**

The paper focuses on indirect answers, a specific type of response to yes-no questions. The authors argue that these types of answers are more commonly observed in human language. The study primarily addresses non-English indirect answers, which have been less explored in previous research. The authors curate new datasets of indirect questions using distant supervision based on rules provided by native speakers to identify yes-no questions with indirect answers. Additionally, they introduce a novel annotated benchmark, annotated by native speakers for Hindi, Korean, Chinese, Bangla, Turkish, Spanish, Nepali, and Persian. Finally, the authors conduct several experiments with multilingual pretrained transformers to assess the task in a supervised setup using their new datasets and benchmark.

**Questions For The Authors:**

A. I may have overlooked this detail, but could the authors elaborate on the segment where indirect answers are identified during the distant supervision phase? While the rules for direct answers and yes-no questions are provided, the methodology for pinpointing non-direct answers to those questions remains unclear. Could you please clarify this aspect?

**Reasons To Accept:**

The paper's primary contribution is the dataset, introducing a novel expertise level to multilingual QA datasets. The authors have evidently undertaken comprehensive work to devise potential rules for the distant supervision phase across multiple languages. This process offers replicability potential for other languages not addressed in the paper, promising value for the broader multilingual QA community.

In my view, the authors' approach to collecting unannotated corpora for each language for the distant supervision phase is commendable. It serves as an exemplary method for constructing intricate datasets and tasks using available data. Well done.

**Reasons To Reject:**

The manuscript could benefit from further refinement in terms of writing quality to meet the standard expectations of a scientific publication. Some statements appear to be general and would be strengthened with additional citations. There are instances where the wording could be more formalized or where unconventional terminology is used. A subsequent round of proofreading and editing might help enhance the presentation of the content.



**Reproducibility:**

3: Could reproduce the results with some difficulty. The settings of parameters are underspecified or subjectively determined; the training/evaluation data are not widely available.

**Reviewer Confidence:**

4: Quite sure. I tried to check the important points carefully. It's unlikely, though conceivable, that I missed something that should affect my ratings.

**Typos Grammar Style And Presentation Improvements:**

The "Related Work" section appears overly broad. Specifically, the paragraph on question-answering seems to merely cite general publications about QA without any clear relevance to your specific work. How does this enhance the reader's understanding? Similarly, the portion discussing multilingual pretraining appears generic. It would be beneficial to the audience to delve deeper into indirect answers and to provide more information about multilingual benchmarks for QA, especially those curated similarly to your approach.

I encountered certain phrases that seemed awkward, including "We build multilingual models" and "computational efforts to date are limited to working with English". For enhanced clarity and formality, consider revising these to sound more precise. This would bolster the reader's confidence in your work. For instance, you might say, "We have developed multilingual models," or "Previous efforts have predominantly focused on English."

---

> ### Author Rebuttal · Authors · 2023-08-28
>
> Thank you for the thoughtful review. We truly appreciate your comments.
>
> **Questions for The Authors**
> - **RE: I may have overlooked this detail, but could the authors elaborate on the segment where indirect answers are identified during the distant supervision phase?**
>    - We develop rules to identify yes-no questions with direct answers (l. 255). Answers that are not identified as direct are considered candidate indirect answers. Note that the annotation process to create the benchmarks in each language includes a validation step to check whether the candidate indirect answers are actually indirect. This validation step takes place before annotators assign the interpretation to the indirect answers. Based on this question, we will add these details in the extra page if accepted.
>
> **Typos Grammar Style and Presentations Improvements**
> - **RE: The "Related Work" section appears overly broad.**
>     - Thank you for the suggestion, we agree after reading it again. We will include multilingual QA (outside yes-no questions) as suggested, including the following papers:
>        - Learning Disentangled Semantic Representations for Zero-Shot Cross-Lingual Transfer in Multilingual Machine Reading Comprehension (Wu et al., ACL 2022)
>        - Synthetic Data Augmentation for Zero-Shot Cross-Lingual Question Answering (Riabi et al., EMNLP 2021)
>        - Automatic Spanish Translation of SQuAD Dataset for Multi-lingual Question Answering (Carrino et al., LREC 2020)
>    - Regarding the generic QA references, we will rephrase to emphasize the difference in the tasks (finding answers or interpreting the answers).
> - **RE: I encountered certain phrases that seemed awkward.**
>    - Thank you for the suggested improvements. We will update the paper and recruit a native speaker to find more improvements.

---

### Meta-Review · Area_Chair_4BCA · 2023-09-19

**Recommendation:** 3

**Metareview:**

The authors introduce a new dataset for interpreting indirect yes/no questions. A training set was constructed using distant supervision based on rules covering in 8 languages, and small test sets of human-annotated examples were created for 3 languages. They also run experiments to provide some baseline results using the data.

The reviewers are in agreement that the data collection seems reasonable and useful for the task, but they did not find it overall exciting beyond that.

---

### Decision · Program_Chairs · 2023-10-07

**Decision:**

Accept-Findings

**Comment:**

The authors introduce a new dataset for interpreting indirect yes/no questions. A training set was constructed using distant supervision based on rules covering in 8 languages, and small test sets of human-annotated examples were created for 3 languages. They also run experiments to provide some baseline results using the data.

The reviewers are in agreement that the data collection seems reasonable and useful for the task, but they did not find it overall exciting beyond that.